# Analysis of Weakening Law and Stability of Sliding Zone Soil in Thrust-Load-Induced Accumulation Landslides Triggered by Rainfall Infiltration

**Zhou Zhou** [1,2], **Junhui Shen** [1,2,*], **Song Tang** [1,2], **Weifeng Duan** [3], **Jingyong Wang** [3], **Richang Yang** [3], **Shengzhe Zheng** [1,2] and **Fulin Guo** [1,2]

[1] State Key Laboratory of Geohazard Prevention and Geoenvironment Protection, Chengdu University of Technology, Chengdu 610059, China; zhouzhou@stu.cdut.edu.cn (Z.Z.); tangsong@stu.cdut.edu.cn (S.T.); zhengshengzhe@stu.cdut.edu.cn (S.Z.); guofulin@stu.cdut.edu.cn (F.G.)

[2] College of Environment & Civil Engineering, Chengdu University of Technology, Chengdu 610059, China

[3] Power China Huadong Engineering Corporation Limited, Hangzhou 310000, China; duan_wf@ecidi.com (W.D.); wang_jy@ecidi.com (J.W.); yang_rc@ecidi.com (R.Y.)

[*] Correspondence: shenjunhui@cdut.cn; Tel.: +86-137-3490-8625

**Abstract:** This study investigated the weakening model, law of mechanics parameters, and stability of the sliding zone soil associated with thrust-load-induced accumulation landslides triggered by rainfall infiltration. The spatial and morphological characteristics and rule of the sapping process were analyzed, considering the constitutive equation of the sliding zone soil, in order to establish a state curve equation for the weakening coefficient of sliding zone soil based on the "S"-shaped curve. Moreover, a formula for calculating slope stability with this failure mode was derived and applied to calculate the stability of a deformation body in Danbo reservoir, China. The results show that the sliding zone in this type of landslide exhibits steep upward and slow downward trends, and affected by rainfall infiltration, its failure develops gradually from the trailing edge to the front edge. In the constitutive equation, the weakening of soil mechanical parameters is manifested as the weakening of shear stiffness, while the "S"-shaped curve of the weakening coefficient reflects the spatial characteristics of the weakening process. The main factors affecting the accuracy of the slope stability calculation are the values of model parameters and assessment of the development characteristics and weakening stage of the sliding zone.

**Keywords:** rainfall infiltration; accumulation landslide; sliding zone soil; weakening coefficient; "S"-shaped curve; weakening law; slope stability

## 1. Introduction

Accumulation landslides refer to a type of landslide associated with Quaternary and modern loose accumulation slopes [1], and they account for a large proportion of global landslides [2]. Such accumulation slopes mainly develop in deep valleys with complex geological settings [3,4]. Accumulation bodies are a product of interactions between internal and external geological processes [5–8], and the formation of an accumulation body has apparent spatiotemporal characteristics, controlled by the comprehensive actions of static and dynamic factors. Therefore, accumulation landslides usually have the following characteristics: (1) The internal cause of the occurrence of accumulation landslides is mainly controlled by the weak surface of the original slope. The sliding surface is usually the base interface or various relatively weak interfaces (belts) in the accumulation body. (2) Controlled by the weak surface, they usually exhibit creep–slip characteristics. (3) Rainfall infiltration into the slope, which further weakens the surface strength parameters of the rock-soil mass, is the main factor inducing this type of landslide. (4) The strength properties of the weak surface of the rock-soil mass gradually weaken with time, and the landslide presents the characteristics of gradual deformation and failure at the macro level.

Regarding the formation mechanism of such landslides and considering the above-mentioned characteristics of accumulation landslides, the current research focus is on the water-soil interaction mechanism after rainfall infiltration: (1) Water-soil mechanical reactions in the accumulation body caused by rainfall infiltration are mainly characterized by changes in hydrostatic and hydrodynamic pressure due to saturated seepage [9–17] and matric suction changes due to unsaturated seepage [18–23]. The changes in hydrostatic pressure are reflected by increases in pore water pressure in the rock-soil mass, which reduce the effective stress and shear strength on the potential sliding surface. The changes in hydrodynamic pressure are reflected by the transport of fine-grained materials in the rock-soil mass to the lower part of the slope. An increase in pore water pressure will decrease the matric suction of the rock-soil mass, thus reducing its shear strength. (2) Water-soil physical and chemical reactions caused by rainfall infiltration [24,25] are mainly characterized by changes in the chemical composition or structure [26–30] and the weakening of mechanical properties [31–33]. Changes in the chemical composition or structure are mainly attributable to large-scale ion exchanges between rainfall infiltration and sliding zone media, and the attenuation of rock-soil mass mechanical parameters is mainly attributable to the action of acidic and alkaline water. (3) Based on unsaturated soil mechanics, saturated–unsaturated constitutive relationship models, and other theories, the characteristics of changes in the slope seepage field, influence of stability, and sliding mechanism under various rainfall conditions have been mathematically analyzed [34–38]. Although previous studies have provided deep insight into the mechanism of accumulation landslides, the following shortcomings remain to be addressed:

(1) The slope body structure of accumulation slopes has not been taken into account-the spatial structure of the weak surface controls the rainfall infiltration process and landslide failure mode [39];

(2) Research results on the mechanisms of water-soil interactions after rainfall infiltration are not universal because some landslides have extremely unique characteristics [9–33];

(3) The characteristics of gradual creep failure in accumulation landslides have not been considered [40];

(4) After rainfall infiltration, the mechanical properties of accumulation slopes exhibit gradual weakening, however the geotechnical physical parameters considered in the calculation of slope stability are all fixed values. Therefore, the spatiotemporal variability of the mechanical parameters is neglected [41,42];

(5) The traditional stability calculation method cannot fully reflect the characteristics of shear stress and displacement changes in the sliding zone [43].

Considering the characteristics of thrust-load-induced accumulation landslides triggered by rainfall infiltration and the limitations of the current analysis methods [44,45], this paper proposes a novel approach toward studying the weakening law of sliding zone soil and determining the stability of such landslides. In this study, the constitutive equation for the sliding zone soil and the "S"-shaped curve are applied to establish a state curve equation for the weakening coefficient of sliding zone soil.

## 2. Failure Mode of Thrust-Load-Induced Accumulation Landslides Triggered by Rainfall Infiltration

In the traditional classification of accumulation landslides, the deformation and failure of slopes can be divided into two basic types according to the mechanical mechanism of deformation and failure: retrogressive landslides (backward type) and thrust-load-induced landslides (forward type) (Figure 1). This classification emphasizes the deformation force and development trend of slope deformation at the macro level and has the advantages of ease of visualization and simple and direct identification (Figure 1) [44,46].

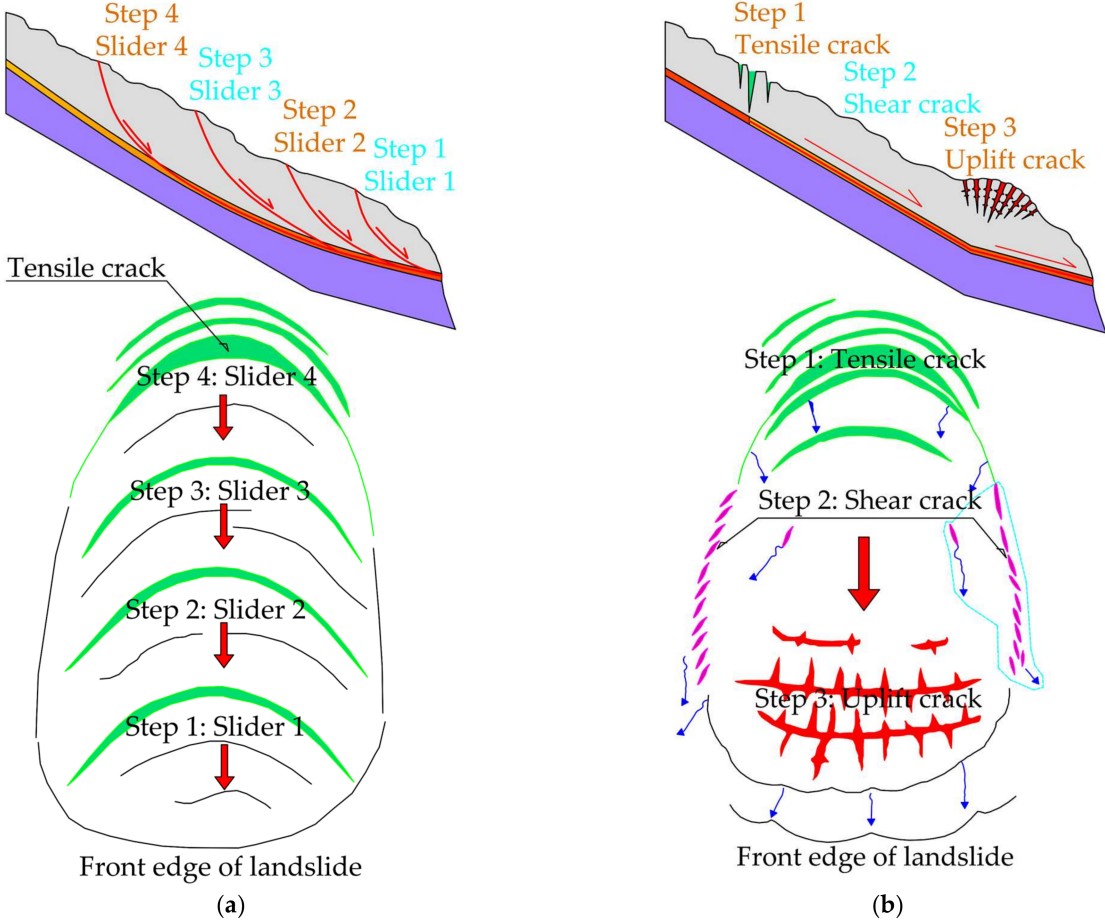

**Figure 1.** Traditional classification of accumulation landslides. (**a**) Schematic diagram of the classic structural sections of retrogressive accumulative landslides and the staged matching system for surface cracks. (**b**) Schematic diagram of the classic structural section of thrust-load-induced accumulation landslides and the staged matching system for surface cracks.

Retrogressive accumulation landslides are characterized by relatively gentle dip angles of the sliding surface, with generally abundant free space at the leading edge (usually attributable to water erosion and cuts, reservoir water level changes, or engineering excavation and other factors), while deformation of the leading edge is usually initiated by the action of gravity. After local sliding failure of the rock-soil mass at the leading edge, a new free surface is formed, which leads to local sliding failure of the rock-soil mass adjacent to the leading edge, and so on. Macroscopically, it shows a sliding mode of "gradual retreat" extending from the front to the back [44,46].

Regarding thrust-load-induced accumulation landslides, the main "force source" is the trailing edge. Usually, rainfall infiltrates the weak zone along the trailing edge and gradually seeps into the leading edge of the slope. The mechanical parameters of the weak zone also gradually weaken from the trailing edge to the leading edge according to the law of rainwater seepage. Macroscopically, the displacement of the trailing edge is greater than that of the leading edge (Figures 1 and 2) [44,45]. The sliding surface of such landslides usually presents the spatial distribution characteristics of a two-stage composite sliding surface with steep upward and slow downward movement [44,45], and their sliding dip angles are usually between 10° and 35° [47–49]. Some examples of such landslides are the Xitan and Bazimen landslides [50,51] in Zigui county, Hubei province, and the Yuhuangge landslide in Wushan county [52].

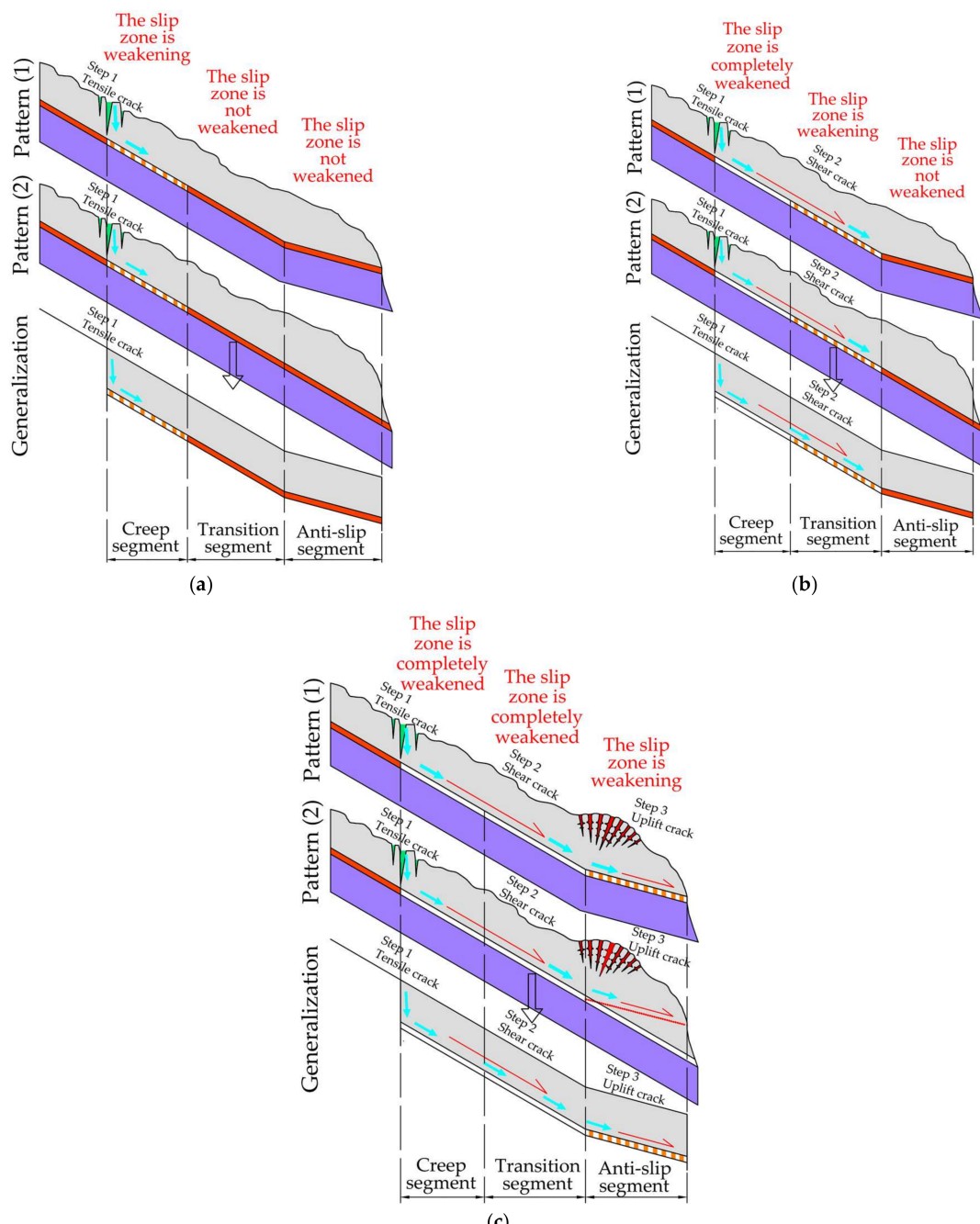

**Figure 2.** Two patterns and generalized models of thrust-load-induced accumulation landslides [44]: (**a**) step 1; (**b**) step 2; (**c**) step 3.

Slope failure is the final stage of the cumulative deformation of a slope [44,45]. According to current research results on thrust-load-induced accumulation landslides triggered by rainfall infiltration [44,45], in terms of the evolution of the sliding zone, the slope usually undergoes three stages: the creep segment, transition segment, and anti-slip segment (shear segment) [47–49]. The shear segment is the final stage of the development of landslide creep, and its sliding zone has a relatively gentle dip angle, while the transition segment is a stage between the two (Figures 1 and 2).

Numerous studies have conducted geotechnical and physical tests [47–52] and reported that rainfall infiltration weakens the mechanical properties of most rock and soil masses in slopes. Accordingly, this study considered strain weakening of soil in the sliding

zone and temporarily ignored the influences of the increases of the sliding body weight and sliding force caused by rainfall.

## 3. Weakening Law of the Mechanical Parameters of Sliding Zone Soil of Thrust-Load-Induced Accumulation Landslides Triggered by Rainfall Infiltration

### 3.1. Constitutive Model of Sliding Zone Soil

The weakening characteristics in the sliding zone and the relationship between shear stress and shear displacement can be effectively reflected by combining the negative exponential constitutive equation and the constitutive equation of soil in the sliding zone [43,49,53,54].

$$\tau(u) = G_S \frac{u}{h} exp\left\{ -\left(\frac{u}{u_0}\right)^m \right\} \tag{1}$$

where $\tau(u)$ is the shear stress on the sliding zone soil and $G_S$ is the initial shear modulus of the sliding zone soil. The initial slope of the curve of the governing equation reflects the stiffness of the sliding zone. Here, $u$ is the slip distance of the slide body relative to the slide bed in the sliding direction; $h$ is the thickness of sliding zone soil; $u_0$ is the average strain measure, which is related to the stress level and increases with increasing normal stress and the position of the peak point of the curve of the governing equation; $m$ is the brittleness index, which is positively correlated with the weakening characteristics of the sliding zone soil (Figure 3). In actual landslides, the peak strength displacement of the sliding zone soil also increases with thickness.

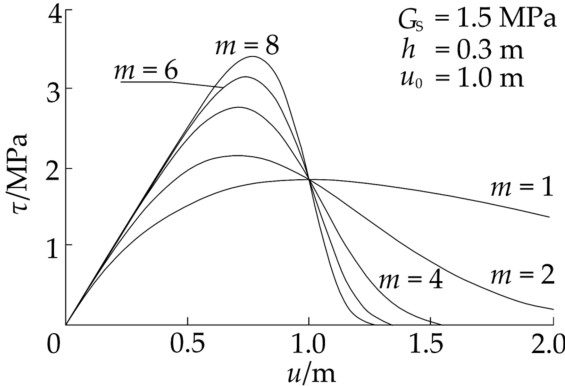

**Figure 3.** Constitutive curves of sliding zone soil for different values of $m$.

Take the first and second derivatives of $u$ in Equation (1):

$$\tau'(u) = \frac{G_S}{h} exp\left\{ -\left(\frac{u}{u_0}\right)^m \right\} \left\{ 1 - m\left(\frac{u}{u_0}\right)^m \right\} \tag{2}$$

$$\tau''(u) = \frac{G_S}{h} exp\left\{ -\left(\frac{u}{u_0}\right)^m \right\} \left\{ -\frac{m}{u_0}\left(\frac{u}{u_0}\right)^{m-1} \right\} \left\{ 1 + m - m\left(\frac{u}{u_0}\right)^m \right\} \tag{3}$$

When $\tau'(u) = 0$ and $\tau''(u) = 0$, the peak intensity $\tau_{max}$ at the curve in Equation (1) and its corresponding displacement $u_f$ and the displacement $u_t$ at the inflection point can be obtained:

$$u_f = u_0 \left(\frac{1}{m}\right)^{\frac{1}{m}} \tag{4}$$

$$\tau_{max} = \frac{G_S}{h} u_0 \left(\frac{1}{m}\right)^{\frac{1}{m}} exp - \frac{1}{m} \tag{5}$$

$$u_t = u_0 \left(\frac{m+1}{m}\right)^{\frac{1}{m}} \tag{6}$$

Through the curve of the Taylor series expansion, Equation (1) before the peak can be expressed as follows:

$$\tau_0(u) \approx \frac{G_S}{h} u \tag{7}$$

Its truncation relative error is:

$$\delta(\tau) = \frac{\Delta\tau}{\tau(u)} = \frac{\tau(u) - \tau_0(u)}{\tau(u)} = \frac{exp\left\{-\left(\frac{u}{u_0}\right)^m\right\} - 1}{exp\left\{-\left(\frac{u}{u_0}\right)^m\right\}} \tag{8}$$

As shown in Figure 4, error curve clusters under different *m* values can be obtained from Equation (6) and the linear generalization accuracy of the constitutive equation curves in the creep stage are positively correlated with the *m* values. As shown in Figure 5, with increased displacement of the sliding zone, it undergoes the softening segment (transition segment) and finally enters the shear segment (anti-slip segment).

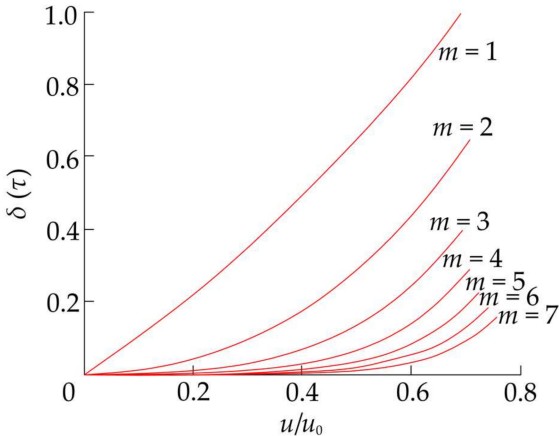

**Figure 4.** Series of error curves for different values of *m*.

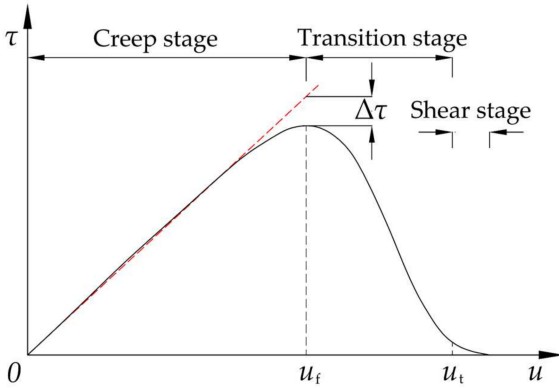

**Figure 5.** Constitutive model curve for sliding zone soil.

*3.2. Weakening Law of the Mechanical Parameters of Sliding Zone Soil*

3.2.1. Basic Weakening Law of Mechanical Parameters

The mechanical parameters of rock and soil materials weakened by external factors (such as rainfall infiltration) are usually reflected by the reduction of cohesion force *c* and internal friction angle *φ*, which can be seen by the weakening of shear stiffness $G_S$ in the constitutive model of the sliding zone soil [43,49,53].

Assuming that the changes of $u_0$ all correspond to normal stress $\sigma$ and the correlation coefficient between them is $P$, we can get:

$$\sigma = P_{u_0} = P\left(\frac{1}{m}\right)^{\frac{-1}{m}} u_f = \xi u_f \tag{9}$$

where $\xi = P(1/m)^{-1/m}$.

Substituting Equation (9) into Equation (5), we get:

$$(u_{0\,i},\, \tau_{\max i}) = \left\{ u_{0\,i},\, \frac{G_s}{h} u_{0\,i} \left(\frac{1}{m}\right)^{\frac{1}{m}} exp\left(-\frac{1}{m}\right) \right\} \tag{10}$$

$$(\sigma_i, \tau_{\max i}) = P_{u_0\,i},\, \frac{G_s}{h} u_{0\,i} \left(\frac{1}{m}\right)^{\frac{1}{m}} exp\left(-\frac{1}{m}\right) \tag{11}$$

$$\left(u_{f\,i},\, \tau_{\max i}\right) = \left\{ \left(\frac{1}{m}\right)^{\frac{1}{m}} u_{0\,i},\, \frac{G_s}{h} u_{0\,i} \left(\frac{1}{m}\right)^{\frac{1}{m}} exp\left(-\frac{1}{m}\right) \right\} \tag{12}$$

According to the basic idea of determining strength through rock-soil mechanics tests, a set of normal shear stress ($\sigma_i$, $\tau_i$) data is fitted by the least squares method to obtain the strength parameters of cohesion $c$ and internal friction angle $\varphi$. Therefore, using the least squares method to fit Equations (11) and (12), $c$ and $\varphi$ can be obtained. Supposing that the slope of the fitted line (equivalent intensity envelope) is $K$ (Figure 6), $\varphi$ can be obtained:

$$\varphi = tan^{-1}\left(\frac{K}{\xi}\right) \tag{13}$$

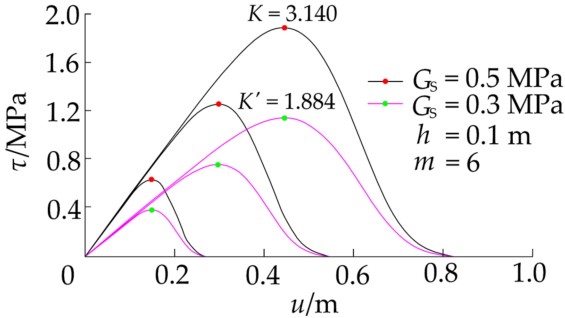

**Figure 6.** Envelope curves of equivalent strength.

Because the $K$ value is related to shear stiffness $G_S$, Equation (13) suggests that the value of $\varphi$ is also related to $G_S$ (Figure 6). For the sliding zone soil of a landslide, suppose that the initial shear stiffness $G_S = 0.5$ Mpa, the brittleness index $m = 6$, and the thickness $h = 0.1$ m; when the shear stiffness decreases to 0.3 Mpa, then the internal friction angle $\varphi'$ of the weakened sliding zone can be determined (Figure 6).

Through further derivation, the internal friction angle $\varphi'$ after the weakening of the sliding zone soil can be expressed as follows:

$$\varphi' = tan^{-1}\left(\frac{K\prime}{K} tan\varphi\right) \tag{14}$$

According to the evolution law of thrust-load-induced accumulation landslides [43–45,47,53], the fundamental cause of its deformation and failure lies in the progressive degradation of the mechanical parameters of sliding zone soil from the back edge to the front edge, and the shear stiffness $G_S$ of the sliding zone soil is directly related to its mechanical parameters

$c$ and $\varphi$. Therefore, the analysis of the weakening law of mechanical parameters in this type of landslide can be generalized to the study of the weakening law of the shear stiffness $G_S$.

### 3.2.2. Weakening Model of Mechanical Parameters

The ratio $k$ of shear stiffness $G'_S$ after weakening and $G_S$ before weakening of sliding zone soil is defined as the weakening coefficient of sliding zone soil [43,53].

$$k = \frac{G'_s}{G_s} \tag{15}$$

For thrust-load-induced accumulation landslides, the $k$ values of the weakening coefficient are not consistent at different positions and at different times. For the original sliding zone soil, $k = 1$; for continuously weakening sliding zone soil, the weakening rate changes at different stages and eventually tends toward an extreme value (Figure 7).

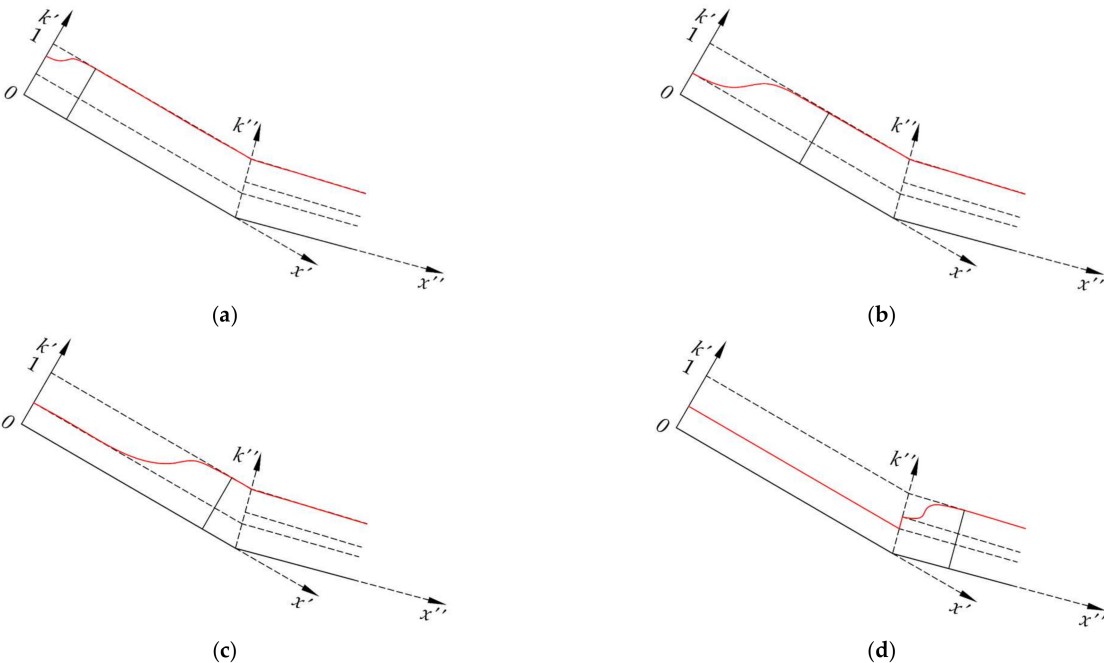

**Figure 7.** Schematic diagram of the "S"-shaped curve of each stage of the weakening of sliding zone soil of thrust-load-induced accumulation landslides triggered by rainfall infiltration: (**a**) the sliding zone begins to weaken locally from the trailing edge; (**b**) a complete "S"-shaped curve is formed; (**c**) the "S"-shaped curve gradually advances towards the front edge; (**d**) it approaches the sliding stage.

During the evolution process, any geological body will go through three periods, ranging from infancy to prime years to decline. Accordingly, an "S"-shaped curve can be used to describe the weakening process of the mechanical parameters of sliding zone soil of thrust-load-induced accumulation landslides [54]. First, the mechanical parameters of sliding zone soil begin to weaken at the trailing edge, as reflected by the $k$ value gradually approaching a certain extreme value and forming an "S"-shaped curve. The weakening phenomenon then gradually advances towards the front part of the landslide. Finally, it advances to the front edge anti-slip segment. When the limit balance between the tangential load and the anti-slip segment is reached, the landslide reaches the critical sliding stage (Figure 7).

### 3.2.3. State Curve Equation for the Weakening Coefficient

According to the definition of the "S"-shaped curve and combined with the weakening characteristics of the sliding zone soil, the state curve equation of the weakening coefficient of the sliding soil can be obtained [43,53]:

$$k(x) = \frac{A}{1 + e^{-c'(x-b)}} + H \tag{16}$$

where $A$ and $H$ are, respectively, the weakening amplitude and weakening limit of the rock-soil mass weakening coefficient at $x$ of the sliding zone and are the attribute parameters of the rock-soil mass itself; $c'$ and $b$ are the form coefficient and the stage representation values of the weakening coefficient state curve (parameters with a specific temporal relationship), respectively.

The "S"-shaped curve is shown in Figure 8; the distance from $H$ to $A$ is $\Delta x$. As the change of the curve always approaches an extreme value infinitely, the tolerance scope of $\Delta x$ needs to be set according to Equation (8) [43,53].

$$\delta(k) = \frac{1-k}{1} = 1 - \frac{1}{1 + e^{-c'\Delta x/2}} = \frac{e^{-c'\Delta x/2}}{1 + e^{-c'\Delta x/2}} = \frac{1}{1 + e^{c'\Delta x/2'}} \tag{17}$$

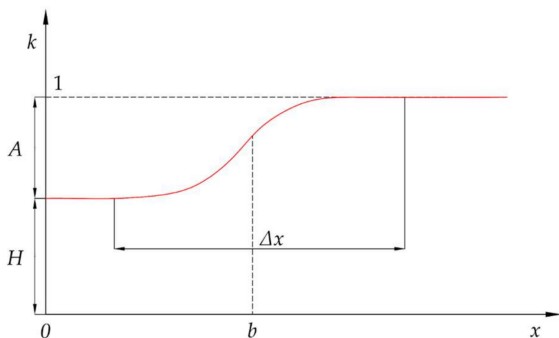

**Figure 8.** The "S"-shaped state curve of the weakening coefficient of the sliding zone soil.

### 4. Stability Calculation of Sliding Zone Soil of Thrust-Load-Induced Accumulation Landslide Triggered by Rainfall Infiltration

Several case studies on landslides have shown [47–53] that the shear stress value of sliding zone soil is closely related to its creep displacement value and the degree of weakening of mechanical parameters. In thrust-load-induced accumulation landslides, the material compositions of the "up-steep" and "down-slow" (The two-stage composite sliding surface with steep upward and slow downward movement) sliding zones exhibit two patterns (Figure 2): (1) For sliding zone soil composed of the same material, the up-steep section and the down-slow section basically have the same composition, for example sliding along the old (ancient) landslide slide zone (Figure 2 Pattern 1). (2) The up-steep section slides along the weak primary surface of the slope, while the down-slow section cuts through the accumulation body; the mechanical properties of the accumulation body are stronger than those of the weak surface (Figure 2 Pattern 2).

Equations (1) and (16) can be used to calculate the anti-shear stress value of a certain position in the sliding zone. Creep slip displacement of sliding zone soil was assumed to be linearly distributed from the back edge to the front edge. The mathematical expression of the anti-slip segment is as follows:

$$u(x) = u(x' + x'') = D\left(1 - \frac{x}{l}\right) = D\left(1 - \frac{x' + x''}{l_1 + l_2}\right) \tag{18}$$

$$\tau'(x,y) = \{k(x') + k(x'')\}\tau(u)$$

$$= \left\langle \left\{ \frac{A_1}{1 + e^{-c_1'(x'-b_1)}} + H_1 \right\} G_{S1} \frac{u}{h} exp\left\{ -\left(\frac{u}{u_0}\right)^{m_1} \right\} \right\rangle$$

$$+ \left\langle \left\{ \frac{A_2}{1 + e^{-c_2'(x''-b_2)}} + H_2 \right\} G_{S2} \frac{u}{h} exp\left\{ -\left(\frac{u}{u_0}\right)^{m_2} \right\} \right\rangle = \tau'(x) = \tau'(x') + \tau'(x'') \qquad (19)$$

$$= \{(x') + k(x'')\}\tau\{u(x)\}$$

where $u(x)$ is displacement at the sliding zone $x$; $D$ is the slide position at the rear edge of the landslide; $x'$ is the coordinate system of the up-steep section; $x''$ is the coordinate system of the down-slow section; $l$ is the apparent length of the sliding zone; and $l_1$ is the apparent length of the up-steep section and $l_2$ is the apparent length of the down-slow section. $A_1$ and $A_2$ and $H_1$ and $H_2$ are the weakening amplitudes and weakening limit values of the rock-soil mass weakening coefficient in the up-steep and down-slow sections, respectively; $c_1'$ and $c_2'$, and $b_1$ and $b_2$ are the shape coefficients and stage representation values of the state curve of the weakening coefficient of rocks and soils in the up-steep and down-slow sections, respectively; $G_{S1}$ and $G_{S2}$ are the shear stiffness of the rock-soil mass in the up-steep and down-slow sections, respectively; $m_1$ and $m_2$ are the brittleness indices of the rock-soil mass in the up-steep and down-slow sections, respectively. If the compositions of sliding zone soil in the up-steep and down-slow sections are the same, $A_1 = A_2$, $H_1 = H_2$, $c_1' = c_2'$, $m_1 = m_2$; if the compositions of sliding zone soil in the up-steep and down-slow sections are differrent, they should be assigned and solved separately (Figure 9).

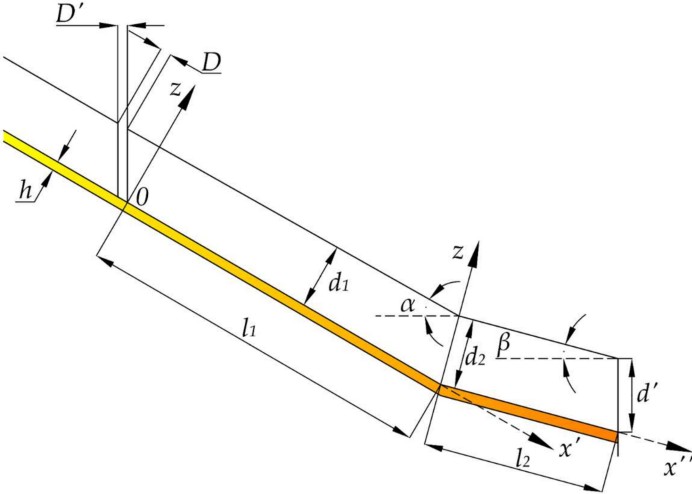

**Figure 9.** Schematic diagram of various morphological parameters reflecting the stability of thrust-load-induced accumulation landslides induced by rainfall infiltration.

Thus, the anti-sliding forces $f_s'$ in the up-steep and $f_s''$ in the down-slow sections of the landslide can be obtained as follows:

$$f_s' = \int_0^{l_1} \tau'(x')dx = \int_0^{l_1} \left\{ \frac{A_1}{1 + e^{-c_1'(x'-b_1)}} + H_1 \right\} G_{S1} \frac{u}{h} exp\left\{ -(\frac{u}{u_0})m_1 \right\} dx \qquad (20)$$

$$f_s'' = \int_0^{l_2} \tau(x'')dx = \int_0^{l_2} \left\{ \frac{A_2}{1 + e^{-c_2'(x''-b_2)}} + H_2 \right\} G_{S2} \frac{u}{h} exp\left\{ -(\frac{u}{u_0})m_2 \right\} dx, \qquad (21)$$

We solve the complex trapezoidal formula:

- Divide the up-steep section and the down-slow section of the sliding zone into equal parts:

$$\Delta h_1 = l_1/n_1, \qquad (22)$$

$$\Delta h_2 = l_2 / n_2, \tag{23}$$

where $\Delta h_1$ and $\Delta h_2$ and $n_1$, and $n_2$ are the calculated step size and equal fractions of the up-steep and down-slow sections, respectively.

- The equidistant nodes can be expressed as follows:

$$x_i' = i\Delta h_1, \tag{24}$$

$$x_j'' = j\Delta h_2, \tag{25}$$

Among them, $x_i'$ and $x_j''$ and $i$ and $j$ are the equidistant value and number of nodes of up-steep and down-slow sections, respectively.

- The sliding resistance force $f_s$ of the sliding zone can be calculated according to two situations: (1) If soil in the up-steep section is not completely weakened, its anti-sliding force $f_s$ is the joint force between the anti-sliding force $f_s'$ of the soil in the up-steep section and the anti-sliding force $f_s''$ of the soil in the down-slow section; (2) If soil in the up-steep section is completely weakened, its sliding resistance force $f_s$ is approximately equal to the sliding resistance force $f_s''$ of the soil in the down-slow section. Under this condition, the calculation equation can be expressed as follows (1):

$$f_s' = \sum_{i=0}^{n_1-1} \frac{\Delta h_1}{2} \left\{ \tau'(x') + \tau'(x_{i+1}') \right\}, \tag{26}$$

$$f_s'' = \sum_{j=0}^{n_2-1} \frac{\Delta h_2}{2} \left\{ \tau'(x_j'') + \tau'(x_{j+1}'') \right\}, \tag{27}$$

$$f_s = \left\{ \left(f_s'\right)^2 + \left(f_s''\right)^2 - 2f_s'f_s''\cos\left(180° - \alpha + \beta\right) \right\}^{1/2}, \tag{28}$$

The calculation equation for case (2) above is as follows:

$$f_s = f_s'' = \sum_{j=0}^{n_2-1} \frac{\Delta h_2}{2} \left\{ \tau'(x_j'') + \tau'(x_{j+1}'') \right\}, \tag{29}$$

Among these, $\tau'(x_i')$, $\tau'\left(x_j''\right)$, $\tau'(x_{i+1}')$, and $\tau'\left(x_{j+1}''\right)$ are the shear stress values at nodes $I$, $j$, $i + 1$, and $j + 1$, respectively.

- The sliding force of the landslide $f_r$ is expressed as follows:

$$f_r' = \rho g d_1 l_1 \sin\alpha, \tag{30}$$

$$f_r'' = \rho g d_2 l_2 \sin\beta, \tag{31}$$

$$f_r = \left\{ \left(f_r'\right)^2 + \left(f_r''\right)^2 - 2f f_r'f_r''\cos\left(180° - \alpha + \beta\right) \right\}^{1/2}, \tag{32}$$

- The slope stability coefficient $F_s$ is expressed as follows:

Situation (1):

$$F_s = f_s / f_r, \tag{33}$$

Situation (2):

$$F_s = f_s''' / f_r = \frac{f_s \sin \beta}{f_r \sin(180° - \beta - \gamma)} = \frac{f_s'' \sin \beta}{f_r \sin(180° - \beta - \gamma)}, \tag{34}$$

$$\gamma = \sin^{-1} \left\{ \frac{f_r'}{f_r} \sin\left(180° - \alpha + \beta\right) \right\}. \tag{35}$$

In the equation, $f_s'''$ is the component of the sliding resistance force of the down-slow section upward on the opposite side of the sliding force $f_r$.

The above medium fractions $n_1$ and $n_2$ are directly proportional to the calculation accuracy of the stability.

## 5. Case Analysis

The Danbo reservoir is located on the right bank (concave bank) in front of the Yangfanggou Hydropower Station in the Yalong River basin, 0.5 km away from the dam. With an elevation range of 2050–2310 m, the slope is mainly composed of multi-stage collapse accumulation. The slope features an ancient landslide accumulation body at an elevation range of 2310–2500 m, with a partial overlying collapse accumulation body. The landslide mass is composed of metamorphic and sandstone block gravel soils and partially of insufficiently disintegrated slate. The sliding zone lies at N 20–60° W/N 16–31° E, and its thickness is 0.6–1.2 m. It is mainly composed of soil mixed with gravel, which is moderately dense in structure and has certain directivity and roundness characteristics (Figure 10) [55,56].

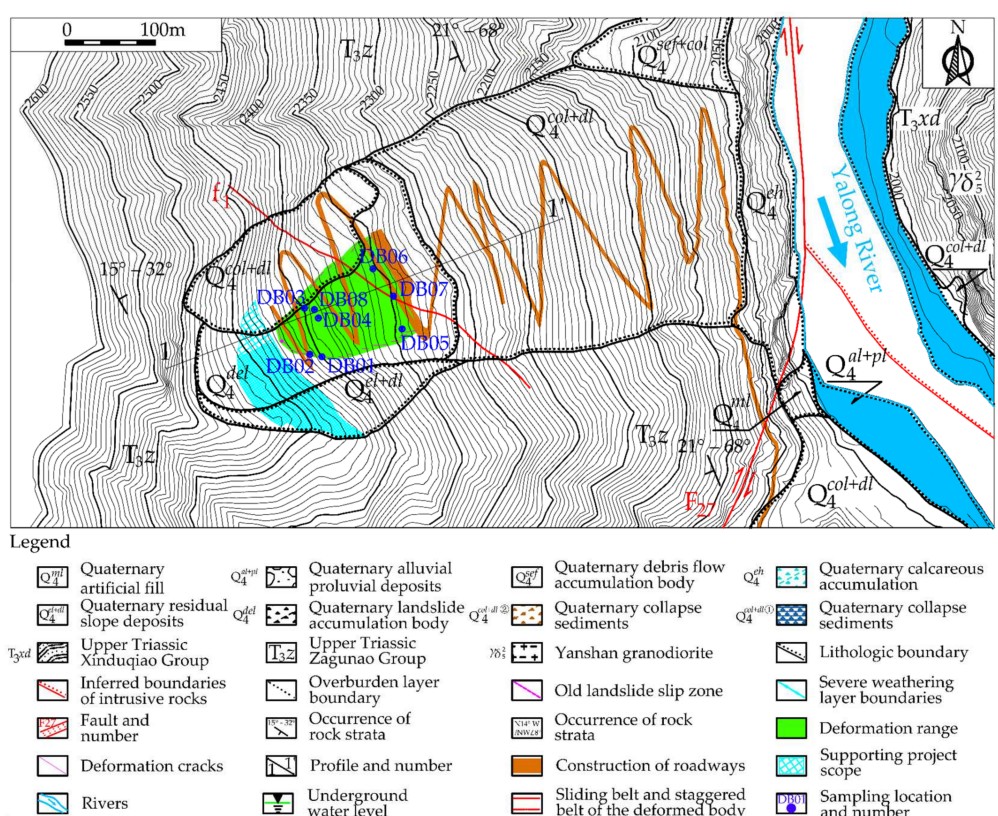

**Figure 10.** Geological engineering plan of Danbo accumulation.

At an elevation of 2340–2420 m, this slope has been experiencing creep deformation since 14 July 2017. However, the deformation tended to be stagnant by 4 November of the same year. The maximum downward dislocation height of the tensile crack at the trailing arc was 1.8 m, while the maximum cumulative displacement in its sliding direction was 2.4 m (Figures 11 and 12). En echelon shear cracks were formed on both sides of the deformation zone, which extend intermittently to the front edge, but no obvious shear outlet could be observed at the front edge. According to the exposed surface during excavation, deformation slides along the ancient landslide slide belt in the elevation range of 2380–2420 m, while the dislocation zone in the elevation range of 2340–2380 m is located in the interior of the ancient landslide. On the whole, there is a two-stage composite sliding zone with "steep-up" and "slow-down" (The two-stage composite sliding surface with steep upward and slow downward movement) characteristics ($31° \rightarrow 16°$) (Figures 11–13). The deformation zone extends from north to south for a length of 145 m and from east to west for a width of 106 m, with a distribution area of $1.4 \times 10^4$ m$^2$, an average thickness of 10 m, and a volume of approximately $15 \times 10^4$ m$^3$. The total length of the slip zone and dislocation zone is 173 m, for which the length of the up-steep section is 60 m with a dip angle of 28–31° and the length of the down-slow section is 113 m with a dip angle of 16–28° (Figures 11 and 13). According to the results of a rock-soil mechanics test, the strength parameters of the slip mass (natural moisture content: $\omega$ = 9.42–12.15%, $c$ = 26 kPa, $\varphi$ = 33.5°; saturated moisture content: $\omega$ = 16.56–17.63%, $c$ = 14 kPa, $\varphi$ = 28.1°) were significantly stronger than those of the slip surface (natural moisture content: $\omega$ = 10.07–16.71%, $c$ = 28 kPa, $\varphi$ = 22°; saturated moisture content: $\omega$ = 18.19–18.66%, $c$ = 13 kPa, $\varphi$ = 14.5°). There is a clear correlation between the ground displacement monitoring results during slope deformation and rainfall events. The cumulative displacement curves for each monitoring point in the deformation area show step-type development, with 4–5 steps for the flooding season. The step-type development of the accumulative displacement curve indicates large early rainfall, whereas a flat accumulative displacement curve indicates small or no early rainfall. Moreover, the occurrence time of the accumulative displacement step also exhibits uniform hysteresis with rainfall time, generally lagging by 1 to 3 days (Figures 11 and 12). Numerical simulation tests of the characteristics of the seepage field change (rainfall data: rainfall monitoring data covering 25 days before deformation; model prototype: Danbo accumulation body 1–1′ geological engineering profile) revealed that the saturated area inside the slope first appeared on the trailing edge of the deformed body. This gradually developed from the trailing edge to the leading edge along the old sliding zone and finally formed a saturated area matching the actual sliding zone (Figure 14). In summary, the deformation of this slope is mainly induced by the weakening of the mechanical parameters of the sliding zone soil due to rainfall infiltration. Rainfall infiltrates the slope body along the excavation face of the back edge and develops along the front edge of the sliding zone and the dislocation zone, gradually weakening the mechanical parameters of the sliding zone and the dislocation zone, which in turn leads to progressive deformation at the front edge (Figures 10, 12 and 13) [55,56].

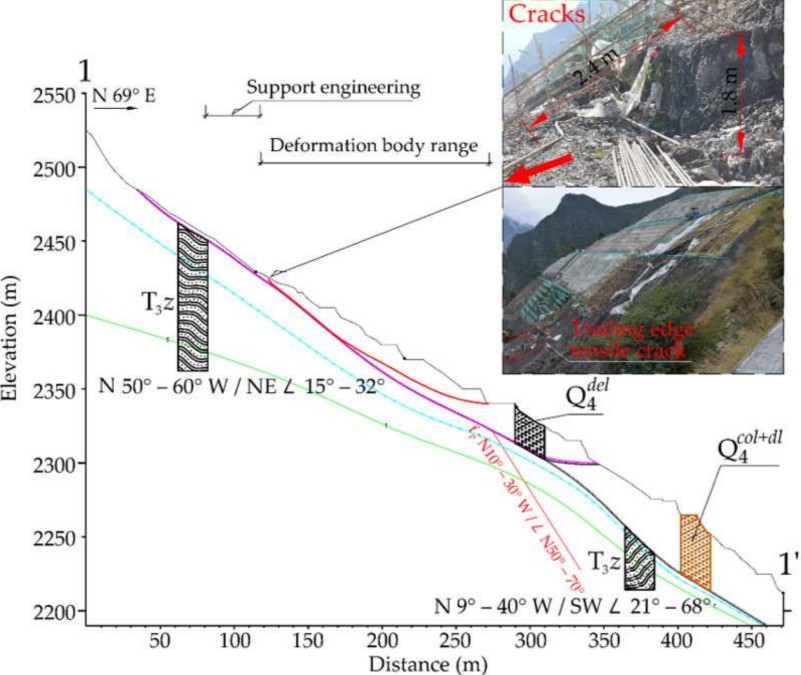

**Figure 11.** Geological engineering profile of Danbo accumulation (photograph taken on 4 November 2017).

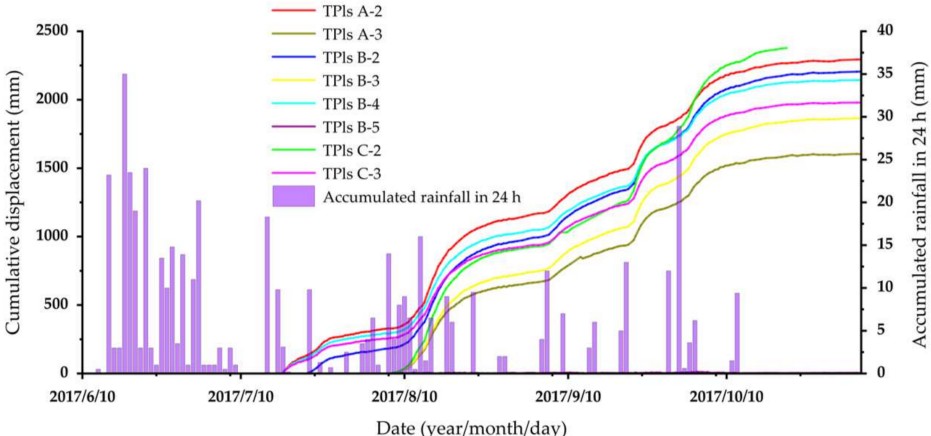

**Figure 12.** Correlation analysis between monitoring results for surface displacement and rainfall.

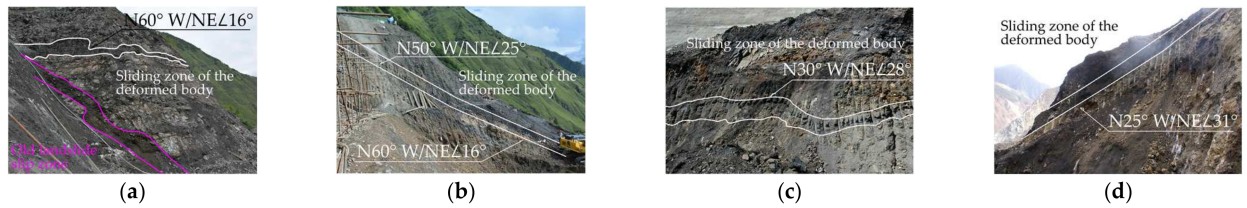

**Figure 13.** Sliding zone characteristics of a deformation body in Danbo reservoir: (**a**) Slip zone at 2330 m elevation; (**b**) Slip zone at 2340–2370 m elevation; (**c**) Slip zone at 2350 m elevation; (**d**) Slip zone at 2410 m elevation.

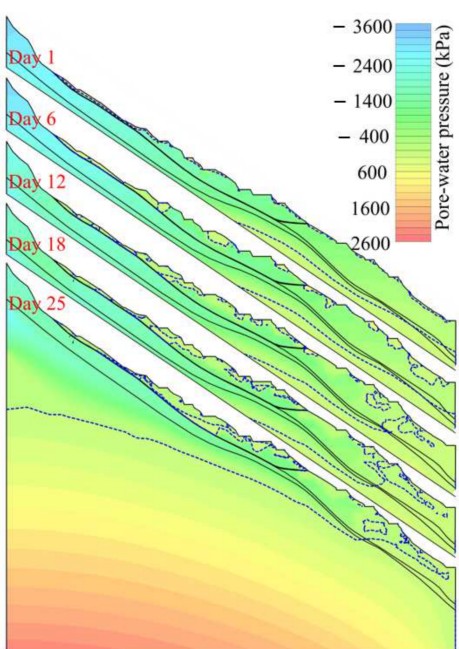

**Figure 14.** Simulation results for characteristic variation in the seepage field.

Based on the above understanding, the model of the weakening of the mechanical parameters was generalized. The parameters were selected according to the test results of the physical and mechanical parameters of rock-soil masses (Table 1) [55,56].

**Table 1.** Parameter values for the slope stability calculation model.

| Steep Upper Sliding Zone Soil | | | | Shallow Lower Sliding Zone Soil | | | |
|---|---|---|---|---|---|---|---|
| **Parameter** | **Value** | **Parameter** | **Value** | **Parameter** | **Value** | **Parameter** | **Value** |
| $A_1$ | 0.44 | $d'/m$ | 10 | $A_2$ | 0.45 | $d'/m$ | 10 |
| $c_1'$ | 0.23 | $d_1/m$ | 8.66 | $c_2'$ | 0.05 | $d_2/m$ | 9.14 |
| $H_1$ | 0.56 | $\alpha/(°)$ | 30 | $H_2$ | 0.55 | $\beta/(°)$ | 24 |
| $G_{S1}/MPa$ | 2.2 | $D'/m$ | 2.08 | $G_{S2}/MPa$ | 2.3 | $D'/m$ | 2.08 |
| $h/m$ | 0.9 | $D/m$ | 2.4 | $h/m$ | 0.9 | $D/m$ | 2.4 |
| $m_1$ | 3 | $\rho_1/(kg \cdot m^{-3})$ | 2360 | $m_2$ | 3 | $\rho_2/(kg \cdot m^{-3})$ | 2260 |
| $u_0/m$ | 0.7 | $l_1/m$ | 60 | $u_0/m$ | 0.7 | $l_2/m$ | 113 |

In Table 1, $G_{S1}$ and $G_{S2}$ are based on the results of on-site rock and soil mechanics tests and are indirectly obtained by combining the Duncan–Zhang bicurve model [57,58]; the values of $A_1$, $A_2$, $H_1$, and $H_2$ are also based on the results of the on-site rock and soil mechanics tests, and are calculated by comprehensively analyzing the drop range of mechanical parameters from natural moisture content to saturated moisture content (cohesion $c$ and internal friction angle $\varphi$); $m_1$ and $m_2$ are empirical values obtained based on the existing sensitivity analysis results [59]; $u_0$ is based on the results of the on-site rock and soil mechanics tests, and its value was obtained from existing research results [43,59]; $\rho_1$ and $\rho_2$ (density) are obtained based on the results of the on-site rock and physical soil tests. The remaining parameters $l_1$, $l_2$, $d'$, $d_1$, $d_2$, $\alpha$, $\beta$, $D'$, $D$ are calculated based on the parameters actually measured on-site combined with the geological generalization model (Figure 10).

Based on the two cases of slip resistance force $f_s$, the stability coefficients of the slope at different evolutionary stages were calculated according to the characteristic values $b_1$ and $b_2$ at different stages. The range of the characterized value $b_1$ of the up-steep section

was 0 m $\leq b_1 < 60$ m, while the range of characterized value $b_2$ of the down-slow section was 0 m $\leq b_2 \leq 113$ m (Figure 15).

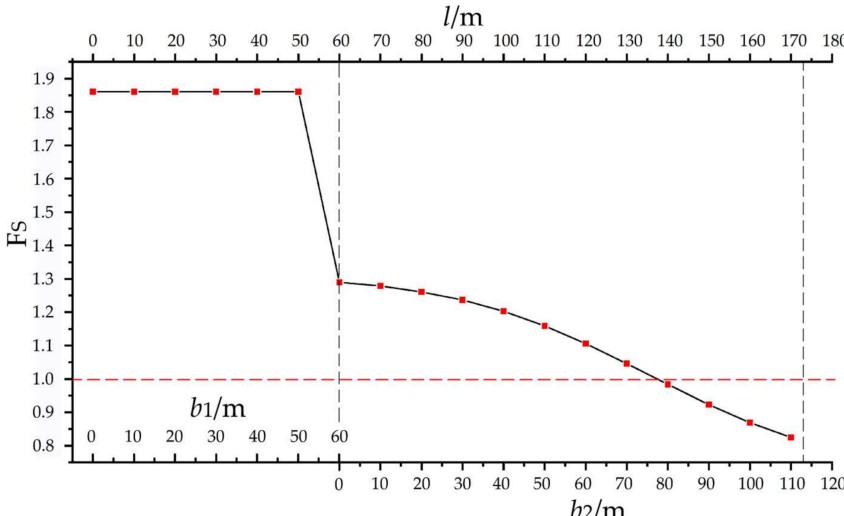

**Figure 15.** Relationship curve between the characteristic values of the Danbo deformation body at different stages and slope stability coefficients.

The stability coefficient of the Danbo deformation body was calculated using MATLAB ($b_1$ to $b_2$). When the mechanical parameters of the sliding zone soil presented an "S"-shaped curve with weakening at the front edge, the change curve of the stability coefficient could be divided into three stages. First, before the weakening of the sliding zone soil from the back edge to the end of the up-steep section, the slope stability coefficient showed slight decreases (magnitude $10^4$). Subsequently, the slope stability coefficient decreased from 1.861 to 1.29 at the end of the weakening of the up-steep section and the beginning of the weakening of the down-slow section. Finally, with the weakening of the down-slow section, the $b_2$–$F_s$ curve exhibited an "S" shape with initial easing followed by steepening. The curve reached a critical sliding state at $b_2 = 78.3$ m. The above $b$–$F_s$ curve indicates that the trailing edge was the "force source" of the thrust-load-induced accumulation landslide, while the sliding of the trailing edge creep section and the transition section had a weak influence on its stability. Overall, the decisive factor controlling its failure was the rock-soil strength of the anti-slide section at the front edge (Figure 15).

## 6. Conclusions

In this study, the weakening model, law of mechanics parameters, and stability of the sliding zone soil of thrust-load-induced accumulation landslides were investigated. The main findings can be summarized as follows:

(1) It is concluded that the deformation and failure mode of thrust-load-induced accumulation landslides is triggered by rainfall infiltration, and the sliding zone is governed by the spatial morphology law of up-steep and down-slow actions. Moreover, the weakening of the sliding zone soil is gradual from the trailing edge to the front edge, with the three stages of creep, transition, and shear;

(2) The negative exponential equation was introduced into the constitutive equation of the sliding zone soil and combined with the constitutive curve. The relationship between shear stress and shear strain in the weakening process of the sliding zone soil caused by rainfall infiltration was expounded in detail from the two aspects of time and space;

(3) In the constitutive model, the weakening law of the mechanical parameters of sliding zone soil can be generalized to weakening shear stiffness. On this basis, the concept of the weakening coefficient is put forward and the state curve equation of the weakening

coefficient of sliding zone soil is established by combining the "S"-shaped curve of the evolution of the geological body;

(4) According to the level of similarity of the material composition of the up-steep section and the down-slow section of sliding zone soil in thrust-load-induced accumulation landslides induced by rainfall infiltration, the two failure modes were further divided and a formula for calculating the stability of such slopes under the two conditions of incomplete weakening of the up-steep section and complete weakening of the up-steep section was derived;

(5) Taking a deformation body in the Danbo reservoir (thrust-load-induced accumulation landslide) as a geological prototype, in which the main factor controlling the landslide is the weakening of mechanical parameters due to rainfall infiltration into the sliding zone, the derived formula was applied and the relationship curve between the characteristic values of $b_1$ and $b_2$ at different weakening stages of the sliding zone soil and slope stability was established. The results reflect the progressive failure characteristics of thrust-load-induced accumulation landslides to a certain extent.

**Author Contributions:** Z.Z., conceptualization, methodology, formal analysis, and writing—original draft preparation. J.S., validation. S.T., and F.G., investigation. W.D., J.W., and R.Y., data curation. S.Z., software and calculation. All authors have read and agree to the published version of the manuscript.

**Funding:** This study was funded by the Natural Science Foundation of China (Grant Nos. 41572308 and 41977226) and Power China Huadong Engineering Corporation Limited.

**Institutional Review Board Statement:** The study does not require ethical approval.

**Informed Consent Statement:** Informed consent was obtained from all subjects involved in the study.

**Data Availability Statement:** The data presented in this study are contained within the article and are available on request from the corresponding author.

**Acknowledgments:** Thanks to the numerous scientists who provided additional information from their studies. The authors also thank anonymous reviewers, editor-in-chief and associate editor for their suggestions on the quality improvement of the present paper.

**Conflicts of Interest:** The authors declare no conflict of interest.

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
