# Peer review of "Analysis of Weakening Law and Stability of Sliding Zone Soil in Thrust-Load-Induced Accumulation Landslides Triggered by Rainfall Infiltration"

_water, doi:10.3390/w13040466_

Round 1
Reviewer 1 Report
The authors implement a semi-empirical approach to predict the occurrence of thrust-load-induced accumulation landslides triggered by rainfall infiltration. The approach includes an “infinite-type” slope stability model that evaluates the stability of a composite failure surface with a steeper upper slope and shallower lower slope. The failure surfaces is assumed along a weak zone, with the strength of the zone being specified by a strain-softening material. The strength distribution along the weak zone is calculated by applying the strain-softening rule to assumed deformations along the weak zone that are based on the field observations. The effects of rainfall-infiltration on slope stability are evaluated by applying a weakening rule that are fitted to observations (e.g., slope failure). Factor of safety is finally calculated as a ratio of resisting over the driving forces.
Although the authors implement an elaborate model, the review considers that the study has several major weaknesses. The main limitations involve the development of a slope stability model that aims to predict rainfall-induced landslides without explicitly modelling the rainfall infiltration process. Instead, a weakening rule is implemented to account for strength reduction resulting from water infiltration. It is not clear from the study what is meant by weakening of soil strength parameters due to rainfall infiltration. Furthermore, the model features a strain-softening model for shear strength based on shear stiffness. However, the strains are not explicitly calculated, but they are assumed based on field observations. More detailed comments are provided below.
Comments:
- Line 43: “Rainfall infiltration into the slope, which further weakens the surface mechanical parameters of the rock soil mass..”. What is meant by weakening? Reduction in soil strength due to loss of suction and increasing pore pressures?
- Line 49: suggest to remove “domestic and foreign”
- Lines 60 to 69: Several limitations of existing studies were mentioned. Please provide references to support these statements.
- Line 75: Define “thrust-load”.
- Line 71: Could not access reference 39, please provide alternative references.
- Line 90: I tried accessing references 40-45 to read more on the background for this study, but could not access them.
- Line 91: “rainfall infiltration weakens the mechanical properties of most rock and soil masses in soil. Accordingly, this study considered strain weakening of soil in the sliding zone.” Strain weakening is characteristic of certain soil types that does not necessarily need to be linked to rainfall infiltration. Please explain what is meant by weakening and how it is related to rainfall infiltration.
- 1: Why does the model not include residual strength?
- Line 106: Explain u0
- Line 110: Derivative with respect to u
- Line 113: Explain the statement in more detail.
- Line 123: Weakening of soil strength parameters such as cohesion and friction angle are embodied in the weakening of shear stiffness. The relation is not stated and the reason for selecting shear stiffness to model weakening of strength is not clear.
- Line 130: Slope K. Please explain.
- 13: provide reference.
- Line 218: What does completely weakened mean?
- 30: Is dry or saturated density used?
- 34: Explain the expression.
- Line 259: What does natural mean? What is the corresponding water content?
Author Response
Dear Reviewer: Greeting! We would like to thank you for your response. We have carefully considered your comments on our manuscript and substantially revised the original manuscript. These revisions are presented in detail below. (1) Your opinion: “Line 43: “Rainfall infiltration into the slope, which further weakens the surface mechanical parameters of the rock soil mass.”. What is meant by weakening? Reduction in soil strength due to loss of suction and increasing pore pressures?”. Reply to this question: The expression of this place has been modified by changing "mechanical parameter" into "strength parameter". The weakening of rock-soil mass here refers to the decrease of c and φ value of strength parameter caused by water-soil interaction after rainfall infiltration into slope. (2) Your opinion: “Line 49: suggest to remove “domestic and foreign””. Reply to this question: Modification has been completed. (3) Your opinion: “Lines 60 to 69: Several limitations of existing studies were mentioned. Please provide references to support these statements.”. Reply to this question: References have been added. (4) Your opinion: “Line 75: Define “thrust-load””. Reply to this question: In the first sentence of the first paragraph of "2. Failure mode of Thrust Load-Induced Accumulation Landslide Triggered by Rainfall Infiltration ", " thrust load-induced accumulation landslide triggered by rainfall infiltration " has been redefined. (5) Your opinion: “Line 71: Could not access reference 39, please provide alternative references”. Reply to this question: The PDF version of this reference has been downloaded and uploaded to the reviewer as an attachment, and a reference has been added here as a supplement. (6) Your opinion: “Line 90: I tried accessing references 40-45 to read more on the background for this study, but could not access them.”. Reply to this question: The PDF version of these references has been downloaded and uploaded to the reviewer as attachments. (7) Your opinion: “Line 91: “rainfall infiltration weakens the mechanical properties of most rock and soil masses in soil. Accordingly, this study considered strain weakening of soil in the sliding zone.” Strain weakening is characteristic of certain soil types that does not necessarily need to be linked to rainfall infiltration. Please explain what is meant by weakening and how it is related to rainfall infiltration.”. Reply to this question: 1) " thrust load-induced accumulation landslide triggered by rainfall infiltration " has been redefined and the mechanism of rock-soil mass strength parameter weakening caused by rainfall infiltration have been added in "1. Introduction" and "2. Failure Mode of Thrust Load-Induced Accumulation Landslide Triggered by Rainfall Infiltration ". 2) Strain weakening is a feature of many rock-soil masses, and it does not necessarily require rainfall infiltration to occur. However, the research object of this paper is a thrust load-induced accumulation landslide, this type of landslide usually occurs because rainfall infiltrates the slope from the trailing edge to accelerate the strain weakening of the rock-soil mass. (8) Your opinion: “Why does the model not include residual strength?”. Reply to this question: The revised Figure 2-5 defines in detail the meaning and generalization basis of the parameters involved in "3. Weakening Law of the Mechanical Parameters of Sliding Zone Soil of Thrust Load-Induced Accumulation Landslides Triggered by Rainfall Infiltration" in this paper. Since this study is based on the geological generalization model of the thrust load-induced accumulation landslide triggered by rainfall infiltration, the author needs to highlight the key points and ignore the influence of some secondary factors in the generalization model for the purpose of the study. (9) Your opinion: “Line 106: Explain u0”;“Line 106: Explain u0”;“Line 110: Derivative with respect to u”; “Line 113: Explain the statement in more detail.”; “Line 123: Weakening of soil strength parameters such as cohesion and friction angle are embodied in the weakening of shear stiffness. The relation is not stated and the reason for selecting shear stiffness to model weakening of strength is not clear.”; “Line 130: Slope K. Please explain.”. Reply to this question: The author re-states the above problems in detail in "3. Weakening Law of the Mechanical Parameters of Sliding Zone Soil of Thrust Load-Induced Accumulation Landslides Triggered by Rainfall Infiltration" and adds Figure 2-5. (10) Your opinion: “Line 218: What does completely weakened mean?”. Reply to this question: The author takes into account the two-stage sliding surface morphology of the thrust load-induced accumulation landslide triggered by rainfall infiltration. Complete weakening means that the up-steep section of the sliding zone has completely gone through the three stages of "Creep stage→Transition stage→Shear stage" described in "2. Failure Mode of Thrust Load-Induced Accumulation Landslide Triggered by Rainfall Infiltration". (11) Your opinion: “Is dry or saturated density used?”; “Line 259: What does natural mean? What is the corresponding water content?”. Reply to this question: 1) "ρ" refers to the density of rock-soil in its natural state. 2) The natural state refers to the state when the researcher obtains the sample from the landslide site. The natural moisture content of the sliding zone soil of the ancient landslide is between 10.07% and 16.71%, and the natural moisture content of the sliding body material is between 9.42% and 12.15 %. Kind regards, Dr. Zhou Zhou State Key Laboratory of Geohazard Prevention and Geoenvironment Protection, Chengdu University of Technology
Reviewer 2 Report
The paper presents a deterministic method to estimate slope stability towards landslides affecting loose materials fomed by accumulation deposits of past landslides.
The work is interesting and reports a robust model to solve slope stability problems towards these complex phenomena. However, it is required to clarify some aspects in the paper and to discuss the achieved results more-in-depth.
Suggested reviews follow:
- In Introduction section, it is important to define also the typical range of velocity of displacements for accumulation landslides
- Explain the reasons why you have chosen these types of costitutive models to explain the weakening along the sliding surface and of the collapse materials
- Add several information on the physical and grain size properties of the unstable mass
- Is there a threshold of rainfall amount above which displacement is characterized by a significant acceleration? Moreover, it is important to indicate the amounts of the different events of acceleration, also according to the rainfall events causing these
- More indications of field monitoring data of displacements (types of sensors, depth of installation, time resolution) are required
- More information on the sensitività analyses carried on to determine m1 and m2 parameters are required
- Discussions of the achieved results are too limited. They could be integrated with: 1) comparison of the developed method with other deterministic methods developed to assess slope stability towards this type of landslides; 2) advantages and limits of the methodology; 3) potential application of the method for hazard analyses
Author Response
Dear Reviewer:
Greeting!
We would like to thank you for your response. We have carefully considered your comments on our manuscript and substantially revised the original manuscript. These revisions are presented in detail below.
(1) About the introduction
Your opinion: “In Introduction section, it is important to define also the typical range of velocity of displacements for accumulation landslides”.
Reply to this question: The research theme of this paper has little correlation with the overall sliding displacement velocity of accumulation landslide. The displacement u in this paper refers to the slip distance of the slide body relative to the slide bed in the sliding direction. The author believes that the range of displacement velocity added in this paper is inconsistent with the theme of this paper, so we hope the reviewer can understand. In addition, in "1. Introduction" and "2. Failure Mode of Thrust Load-Induced Accumulation Landslide Triggered by Rainfall Infiltration", the author added the mechanism of the weakening of rock-soil mass strength parameters caused by rainfall infiltration, the research background, the definition of thrust load-induced accumulation landslide triggered by rainfall infiltration and the expression of methodology for research.
(2) About model selection
Your opinion: “Explain the reasons why you have chosen these types of costitutive models to explain the weakening along the sliding surface and of the collapse materials”.
Reply to this question: 1) the thrust load-induced accumulation landslide triggered by rainfall infiltration has its special slope structure and failure mode, especially the spatial structure of the weak surface or sliding surface. For example: a two-stage composite sliding zone with steep up and slow down, the strength of the sliding zone soil shows a gradual weakening from the trailing edge to the leading edge (gradual creep failure), and whether the material composition of the up-steep section and the down-slow section are the same , whether the soil in up-steep section is completely weakened, etc. Therefore, it is necessary to construct a geological generalization model of such landslides based on the above characteristics. 2) At present, in the research of slope stability calculation, the research results on the weakening of the strength parameters caused by the water-soil interaction involved are not universally applicable, and the rock-soil mass mechanical parameters used in the stability calculation are all fixed value, ignoring the timeliness and spatial variability of its mechanical parameters. In fact, the mechanical parameters of the rock-soil mass after rainfall infiltration slope are a process of gradual damage and weakening. Therefore, the weakening coefficient state curve of the sliding zone soil based on the constitutive equation of the sliding zone soil and the "S" curve is selected in this paper. The equations can generalize the study of the traditional strength parameters c and φ into the study of the weakening law of shear stiffness Gs, and the stability calculation formula finally obtained can also reflect the characteristics of sliding zone shear stress and displacement changes to a certain extent.
(3) About adding sliding zone soil physical properties and more field displacement monitoring data
Your opinion: “Add several information on the physical and grain size properties of the unstable mass”; “ More indications of field monitoring data of displacements (types of sensors, depth of installation, time resolution) are required”.
Reply to this question: 1) The state curve equation of the sliding zone soil weakening coefficient constructed in this paper, the calculation formula of slope stability derived, the physical and mechanical properties of the sliding zone soil and other parameters involved have all been explained in this paper. The author believes that the addition of physical and mechanical parameters that have not yet been involved will be inconsistent with the subject of this paper, and the reviewer are kindly requested to understand. 2) The author also believes that the content such as the field displacement monitoring data added in the case does not belong to the focus of this paper. The monitoring data in "Figure 11 Correlation analysis between monitoring results of surface displacement and rainfall" can effectively reflect the correlation between the landslide and rainfall (“The step type development of the accumulative displacement curve indicates large early rainfall, whereas a flat accumulative displacement curve indicates small or no early rainfall. Moreover, the occurrence time of the accumulative displacement step also exhibits uniform hysteresis with rainfall time, generally lagging by 1 to 3 days.”). In addition, the parameters in the stability calculation formula do not involve the displacement parameters of the landslide, and the reviewer are kindly requested to understand.
(4) About the analysis of correlation between added rainfall and landslide displacement and destruction
Your opinion: “Is there a threshold of rainfall amount above which displacement is characterized by a significant acceleration? Moreover, it is important to indicate the amounts of the different events of acceleration, also according to the rainfall events causing these”.
Reply to this question: Yes, for the study case, there is a correlation between rainfall and the overall displacement of the landslide, which the author has stated in L302-307 (“The step type development of the accumulative displacement curve indicates large early rainfall, whereas a flat accumulative displacement curve indicates small or no early rainfall. Moreover, the occurrence time of the accumulative displacement step also exhibits uniform hysteresis with rainfall time, generally lagging by 1 to 3 days.") and "Figure 11 Correlation analysis between monitoring results of surface displacement and rainfall". If the reviewer considers that the author needs more data to support the above statements, the author may add a table in this paper ("Summary of characteristics of time-cumulative displacement/displacement rate curve for monitoring points in the major deformation area"). However, this will make the article seem not concise enough, hope the reviewers will understand.
(5) Information about adding parameter m
Your opinion: “More information on the sensitività analyses carried on to determine m1 and m2 parameters are required”.
Reply to this question: As for the definition of parameter m, the author has added it in "3. Weakening Law of the Mechanical Parameters of Sliding Zone Soil of Thrust Load-Induced Accumulation Landslides Triggered by Rainfall Infiltration", including textual expressions and Figures 2, 3, and 5; As for the values of m1 and m2 in the case study, the author took the empirical values based on previous sensitivity studies, and quoted relevant articles in the paper ([58]).
(6) About the discussion
Your opinion: “Discussions of the achieved results are too limited. They could be integrated with: 1) comparison of the developed method with other deterministic methods developed to assess slope stability towards this type of landslides; 2) advantages and limits of the methodology; 3) potential application of the method for hazard analyses”.
Reply to this question: 1) As there is currently no special research on the stability calculation of the thrust load-induced accumulation landslide triggered by rainfall infiltration, it is temporarily impossible to compare. In addition, the author used the SEEP/W and SLOPE/W modules (Morgenstern-Price method) based on GeoStudio software to calculate the stability of this case in the relevant research. If add this calculation result for comparison, the author thinks it will make the paper not concise enough and fail to highlight the theme and focus of the research, hope the reviewers will understand. 2) The advantages of this method include: a detailed summary of the special slope structure and failure mode of the thrust load-induced accumulation landslide triggered by rainfall infiltration, and the above characteristics are reflected in the derivation of the stability calculation formula.It avoids the problem of determining the value of rock-soil mass mechanical parameters in traditional stability calculation, and can effectively reflect the timeliness, spatial variability of sliding zone soil mechanical parameters, and the characteristics of sliding zone shear stress and displacement changes. The limitations of this method include: the generalized model can only reflect the main characteristics of such landslide, ignoring the influence of the increase of sliding body weight and sliding force caused by rainfall infiltration. 3) The author in this paper summarizes the special slope structure and failure mode of the thrust load-induced accumulation landslide triggered by rainfall infiltration, and on the basis of using the constitutive equation of the sliding zone soil and the "S" curve to establish the state curve equation of the weakening coefficient of the sliding zone soil, propose a method for the study of weakening laws of the sliding zone soil and stability solution suitable for such landslides, which provides a useful supplement to the theory of landslide prevention.
Kind regards,
Dr. Zhou Zhou
State Key Laboratory of Geohazard Prevention and Geoenvironment Protection, Chengdu University of Technology

Reviewer 3 Report
Dear Authors,
Your paper entitled Analysis of Weakening Law and Stability of Sliding Zone Soil in Thrust Load-Induced Accumulation Landslides Triggered by Rainfall Infiltration (ID: water-1048127) represents a necessity in studying landslides triggered by rainfall infiltration. It is a pretty straightforward paper that goes directly to the point it needs to address, and the rationale behind it is clearly stated in the introduction section. However, it needs a bit more international context (you should correct this aspect), as you refer only to China.
The English of the manuscript is good. At this stage, I recommend Minor Revisions, because there are some more things to be considered and which are mentioned below:
L36: maybe it is better to replace “environment” with “setting”
L58: after [34-38] you need a full stop
You should better define, within the manuscript, the research background and the methodology sections
L280-289: that should be somehow included in the figure 6a; same with L296-298.
Figure 6a and 6b are not mentioned in the main body of the manuscript. Correct that
Making 2 figures out of Figure 6 will give more clarity
Kind regards,
Good luck with the review.
Author Response
Dear Reviewer:
Greeting!
We would like to thank you for your response. We have carefully considered your comments on our manuscript and substantially revised the original manuscript. These revisions are presented in detail below.
(1) Your opinion: “L36: maybe it is better to replace “environment” with “setting””.
Reply to this question: Modification has been completed.
(2) Your opinion: “L58: after [34-38] you need a full stop, You should better define, within the manuscript, the research background and the methodology sections”.
Reply to this question: In "1. Introduction" and "2. Failure Mode of Thrust Load-Induced Accumulation Landslide Triggered by Rainfall Infiltration", the author added the mechanism of the weakening of rock-soil mass strength parameters caused by rainfall infiltration, the research background, the definition of thrust load-induced accumulation landslide triggered by rainfall infiltration and the expression of methodology for research.
(3) Your opinion: “L280-289: that should be somehow included in the figure 6a; same with L296-298.Figure 6a and 6b are not mentioned in the main body of the manuscript. Correct that. Making 2 figures out of Figure 6 will give more clarity”.
Reply to this question: The expressions in Figure 6 and Figure 8 have been modified; Figure 6a and Figure 6b have been modified to Figure 9 and Figure 10 respectively according to the reviewer's requirements, and make the mark in the corresponding position in this paper.
Kind regards,
Dr. Zhou Zhou
State Key Laboratory of Geohazard Prevention and Geoenvironment Protection, Chengdu University of Technology

Round 2
Reviewer 1 Report
After reviewing the revised version of the paper, the reviewer considers that the main limitation of the study remains. The limitation relates to the implementation of a slope stability model that aims to predict rainfall-induced landslides without explicitly modelling the rainfall infiltration process. Instead, a weakening rule is implemented to account for strength reduction resulting from rainfall infiltration based on the displacements measured after a rainfall event. The reviewer considers that the complex processes involved in the initiation of rainfall-induced landslides are oversimplified through this process and that this may overshadow the exact reasons for landslide initiation. Therefore this reviewer considers that the manuscript should not be accepted for publication in the current format. However, the reviewer considers that the proposed approach has potential to be further developed and improved in combination with more explicit modelling of physical processes involved in landslide initiation.
Author Response
Dear Editor and Reviewer: Greeting! We would like to thank you for your response. We have carefully considered your comments on our manuscript and substantially revised the original manuscript. These revisions are presented in detail below. (1)About English language Your opinion: “It is extremely poor, in most point making the text hardly understandable, especially the new parts that have been added in response to the comments raised by the Referees in the first round of review (indeed, this may be the reason of some misunderstanding by Reviewer#1). A thorough review of the language is mandatory, and I suggest you to ask a professional mother tongue to help you in this respect.”. Reply to this question: The author deeply apologize for this, due to time constraints, the last revised manuscript failed to be professionally polished, so lead to the article language expression is difficult to understand, especially the newly revised parts, which may be the reason why Reviewer#1 failed to fully understand the author's true thinking. To this end, the author, relying on a professional polishing agency, has effectively polished the manuscript so that it can truly express the author's ideas. (2)About technical language and methods Your opinion: “There are several mistakes throughout the paper, that make me think that it needs a careful check. I give you an example: when you derive the expression of tau with respect of u, you write that you are deriving u, which is totally wrong. Please, check the correctness of your methods, and how you discuss them in the paper.”. Reply to this question: The reasons for this problem: First, it may be the author lacks professional English polish, for which the author once again apologizes; Second, it may be caused by the difference between Chinese expression and English expression. Regardless of the cause, the author has checked and modified these issues through a professional polishing agency. If there are still similar issues that have not been resolved, the editor is kindly requested to point out that the authors will do his best to modify to meet the publication standards . (3)About scientific content 1 Your opinion: “The novelty of your approach is not clearly stated in the manuscript. Is your "S shaped" curve a novel proposal, or has such model already been proposed in the literature. If so, please explain what is the added value of the application of this model to the landslide case you present, and what kind of general findings can be obttained from it.”. Reply to this question: Regarding the innovative expression of the paper, the author expressed the deficiencies of existing research in Line 68-79 of "1. Introduction" and solved these problems in subsequent research to reflect the innovativeness of this paper. Due to the limited space of the manuscript, it is inconvenient for the author to explain the same expression a second time. This expression may misunderstand the reviewers and editors. To this end, the author will explain the five deficiencies in the study in detail in the cover letter. “1) The slope body structure of the accumulation slope has not been taken into account—the spatial structure of the weak surface controls the rainfall infiltration process and landslide failure mode.” First of all, according to the existing landslide classification method, the author divides accumulation landslides into two types: retro gressive accumulation landslide and thrust load-induced accumulation landslide, and detailed the deformation and failure characteristics of the two types of landslides, especially the deformation and failure characteristics and triggering factors(rainfall infiltration) of the thrust load-induced accumulation landslide studied in this paper(In the “2. Failure Mode of Thrust Load-Induced Accumulation Landslide Triggered by Rainfall Infiltratio” in paragraphs 1-3, Figure 1,Figure 2,Figure 14). Then, from the perspective of the material composition and spatial structure(this is an internal factor that controls the occurrence of the thrust load-induced accumulation landslide, and rainfall infiltration is an external triggering factor) of the rock-soil mass on the slope (soft surface), the spatial distribution characteristics of a two-stage composite sliding surface with steep up and slow down are summarized(Figure 1,Figure 2). Two failure modes are proposed according to whether the material composition of the sliding zone in the down-slow section and the up-steep section is the same(Figure 2), and the influence of the above two modes is considered in the derivation of the stability calculation formula(Line 284-290, Formula 33-35), and the two cases of whether the soil in up-steep section is completely weakened are also considered. The two situations(Line 284-290, Formula 33-35)( Pattern (1)+Situation (1), Pattern (1)+Situation (2), Pattern (2)+Situation (1), Pattern (2)+Situation (2)) in the above two modes(Figure 2) can generally cover all the characteristics of deformation and failure of the thrust load-induced accumulation landslide (triggered by rainfall infiltration). This is not involved in previous studies, so the author believes that this is one of the innovations of this paper. “2) Research results on the mechanism of water–soil interactions after rainfall infiltration are not universal because some landslides have extremely unique characteristics.” At present, there are many research results on the water-soil interaction mechanism caused by rainfall infiltration, which leads to a strong pertinence of each research result. Each landslide has its own characteristics in terms of rainfall infiltration and physical and mechanical properties of rock-soil mass, and the corresponding research results on the water-soil interaction mechanism cannot be fully applicable. Therefore, this paper only briefly describes the mechanism of how the water-soil interaction weakens the mechanical parameters of the sliding zone soil, but directly finds the relationship between the shear stiffness "G" _"S" and the traditional mechanical parameters (Cohesion c, internal friction angle φ),use the weakening coefficient k related to the shear stiffness "G" _"S" to replace the decrease of the mechanical parameters of the sliding zone soil. This method has two advantages: 1. It avoids the influence of multiple mechanical parameters on the calculation, and only uses one parameter for stability calculation, which improves the accuracy of the calculation; 2.This study avoids the most complex and changeable mechanism of water-soil interaction in landslide research and provides a new idea for the study of this kind of landslide. “3) The characteristics of gradual creep failure of accumulation landslides have not been considered. 4) After rainfall infiltration, the mechanical properties of accumulation slopes exhibit gradual weakening, but the geotechnical physical parameters considered in the calculation of slope stability are all fixed values. Therefore, the spatiotemporal variability of the mechanical parameters is neglected.” Rainfall infiltration into the interior of the slope is a gradual process. In the thrust load-induced accumulation landslides triggered by rainfall infiltration, rainfall usually infiltrates the internal sliding zone of the slope along the trailing edge of the landslide, and gradually seepage along the sliding zone to the leading edge (Figure 2,Figure 14), this leads to the weakening of the sliding zone soil is also a gradual process. In the previous studies on the thrust load-induced accumulation landslides, there are few studies on gradual creep, and most of them are qualitative studies; In terms of stability calculation, the sliding zone is usually considered as a whole and each state is given a set of fixed parameters for calculation. It is difficult to summarize the weakening characteristics of the sliding zone as a whole. Therefore, the advantages of this method are as follows: 1. Combine the current theory of gradual creep and push-type accumulation landslide, and apply it to the stability calculation formula on the basis of qualitative analysis and summary. 2. Use the weakening coefficient k related to the shear stiffness "G" _"S" to replace the fixed value of the traditional calculation, and the weakening coefficient k is changing throughout the calculation process, to a certain extent, it reflects the weakening of the mechanical parameters of the sliding zone soil, the characteristic of periodic changes in time, and the characteristics of differences in space. 5) The traditional stability calculation method cannot fully reflect the characteristics of shear stress and displacement changes in the sliding zone. As explained in the previous, in the previous studies on the thrust load-induced accumulation landslides, there are few studies on gradual creep, and most of them are qualitative studies; In terms of stability calculation, the sliding zone is usually considered as a whole and each state is given a set of fixed parameters for calculation. It is difficult to summarize the weakening characteristics of the sliding zone as a whole. In addition, the mechanical parameters (Cohesion c, Internal friction angle φ) given by it are usually obtained from experiments, parameter inversion, and analogy, and it is difficult to reflect the characteristics of shear stress and displacement changes. It is worth noting that the method in this paper is based on the negative exponential constitutive equation and the constitutive equation of sliding zone soil. The equation itself can effectively reflect the relationship between shear stress and shear displacement (Equation 1). Based on the above explanation, the research ideas proposed by the author are innovative:1. First, this paper summarizes the deformation and failure characteristics of the thrust load-induced accumulation landslide. 2. Subsequently, the rock-soil composition and spatial structure characteristics of the sliding zone soil of this type of landslide are summarized, and four modes of deformation and failure of the thrust load-induced accumulation landslide are derived from this. (Pattern (1)+Situation (1), Pattern (1)+Situation (2), Pattern (2)+Situation (1), Pattern (2)+Situation (2)) 3. Then, the qualitative analysis of gradual creep is combined with the constitutive equation, and the weakening coefficient k related to the shear stiffness "G" _"S" is used to replace the mechanical parameters of the sliding zone soil (Cohesion c, Internal friction angle φ), and applying the "S"-shaped curve to establish a state curve equation of the weakening coefficient of sliding zone soil, which can reflect the real situation of the weakening process. 4 Finally, applying the deduced results to the stability calculation, which provides a new idea for slope stability calculation research. (4)About scientific content 2 Your opinion: “The description of the kind of landslides you are studying with your model is unclearly presented. You should be more consistent throughout the paper, using the same terms to define your landslides and the involved materials and, more importantly, you should refer to the terminolgoy of a generally accepted landslide classification (e.g. Varnes, or Hungr). ”. Reply to this question: The author found that by consulting related landslide classification literature (including the scholars listed by the editor). Varnes (classified according to the characteristics of rock and soil mass movement) classifies slope failures into Falls, Topples, Slides, Lateral spreads, Flows, Complex. Hutchinson(classified according to the characteristics of rock and soil mass movement) classifies slope failures into Rebound、Creep、Sagging of mountain slopes、Landslides、Debris movements of flow-like form、Topples、Falls、Complex slope movements. Although their classification is generally accepted by scholars, the starting point and gist of this paper are the mechanical mechanism of deformation and failure of accumulation landslides. Therefore, according to the mechanical mechanism of landslide deformation and failure, the author divides accumulation layer landslides into two types: retro gressive accumulation landslide and thrust load-induced accumulation landslide, and made changes and additions in "2. Failure Mode of Thrust Load-Induced Accumulation Landslide Triggered by Rainfall Infiltratio" (paragraphs 1-3, Figure 1, Figure 2, Figure 14), which introduces the mechanical mechanism of these two types of landslides in detail from both theoretical and practical cases. (5)About scientific content 3 Your opinion: “Overall, it is not clear how you choose the values of your models (degradation of material mechanical properties and evolution of the S-shaped curve) and adapt them to your case study. Back-analysis of field measurements of displacements and rainfall? Literature indications? Without clarifying this, nobody can learn from your study how to apply the presented models.”. Reply to this question: I am very sorry for this, the author's expression in this aspect is somewhat deficient. After the last review, the author has supplemented and perfected the meaning of each parameter in the formula derivation process. If there are still parts that are difficult for readers to understand, the editor is asked to propose and the author continues to modify. In addition, regarding the parameter assignment problem in the case analysis, the author has added this point in the paper (Line 351-368), among which: 1). ” "G" _"S1" and "G" _"S2" were mainly obtained from triaxial test results of sliding zone soil. In the absence of a triaxial test, they can also be indirectly obtained by direct shear test combined with the Duncan-Zhang bicurve model”. The author has listed the references of the solution method. The "G" _"S1" and "G" _"S2" in this paper are calculated indirectly according to this method. 2). “The values of "A" _"1" , "A" _"2" , "H" _"1" , and "H" _"2" are mainly based on the softening coefficient of sliding zone soil”. These parameters are based on on-site rock and soil mechanics test results, and calculated by comprehensively analyzing the drop range of mechanical parameters from natural moisture content to saturated moisture content(Cohesion c, internal friction angle φ), and get the result. 3). "m" _"1" and "m" _"2" are empirical values obtained based on the existing sensitivity analysis results(by consulting the literature "[59]"). 4). "u" _"0" is based on on-site rock and soil mechanics test results, and combined with existing research results give value for it (combine the literature "[43,59]" to obtain comprehensive values). 5). ρ1 and ρ2 are obtained based on on-site rock and soil physical test results. 6). The remaining parameters l_1,l_2,d^’,d_1,d_2,α,β,D^’,D are calculated based on the parameters actually measured on-site and combined with the geological generalization model(Figure 9). (6)Regarding the author’s thanks to the editor and reviewers After two rounds of review, the quality of this paper has been improved due to the editor's suggestions for revision. For this, the author expresses my sincere thanks! There are a total of three reviewers in the first round. Among them, the decisions of Reviewer#1 and Reviewer#2 were major revisions, and the decision of Reviewer#3 was a slight revision. After the first round of revisions, the author was approved by Reviewer#2. I hope the editor can carefully review the paper again this time, and the new opinions put forward will continue to improve the quality of the paper, and the author will continue to actively cooperate with the editor and reviewers. The author apologizes again for the poor English expression before! I hope that mankind can overcome COVID-19, long live the friendship between China and Serbia people! Kind regards, Dr. Zhou Zhou State Key Laboratory of Geohazard Prevention and Geoenvironment Protection, Chengdu University of Technology
Reviewer 2 Report
The Authors replied to all the comments and the manuscript was significantly improved.
Author Response
Dear Editor and Reviewer:
Greeting!
We would like to thank you for your response. We have carefully considered your comments on our manuscript and substantially revised the original manuscript. These revisions are presented in detail below.
(1)About English language
Your opinion: “It is extremely poor, in most point making the text hardly understandable, especially the new parts that have been added in response to the comments raised by the Referees in the first round of review (indeed, this may be the reason of some misunderstanding by Reviewer#1). A thorough review of the language is mandatory, and I suggest you to ask a professional mother tongue to help you in this respect.”.
Reply to this question: The author deeply apologize for this, due to time constraints, the last revised manuscript failed to be professionally polished, so lead to the article language expression is difficult to understand, especially the newly revised parts, which may be the reason why Reviewer#1 failed to fully understand the author's true thinking. To this end, the author, relying on a professional polishing agency, has effectively polished the manuscript so that it can truly express the author's ideas.
(2)About technical language and methods
Your opinion: “There are several mistakes throughout the paper, that make me think that it needs a careful check. I give you an example: when you derive the expression of tau with respect of u, you write that you are deriving u, which is totally wrong. Please, check the correctness of your methods, and how you discuss them in the paper.”.
Reply to this question: The reasons for this problem: First, it may be the author lacks professional English polish, for which the author once again apologizes; Second, it may be caused by the difference between Chinese expression and English expression. Regardless of the cause, the author has checked and modified these issues through a professional polishing agency. If there are still similar issues that have not been resolved, the editor is kindly requested to point out that the authors will do his best to modify to meet the publication standards .
(3)About scientific content 1
Your opinion: “The novelty of your approach is not clearly stated in the manuscript. Is your "S shaped" curve a novel proposal, or has such model already been proposed in the literature. If so, please explain what is the added value of the application of this model to the landslide case you present, and what kind of general findings can be obttained from it.”.
Reply to this question: Regarding the innovative expression of the paper, the author expressed the deficiencies of existing research in Line 68-79 of "1. Introduction" and solved these problems in subsequent research to reflect the innovativeness of this paper. Due to the limited space of the manuscript, it is inconvenient for the author to explain the same expression a second time. This expression may misunderstand the reviewers and editors. To this end, the author will explain the five deficiencies in the study in detail in the cover letter.
“1) The slope body structure of the accumulation slope has not been taken into account—the spatial structure of the weak surface controls the rainfall infiltration process and landslide failure mode.”
First of all, according to the existing landslide classification method, the author divides accumulation landslides into two types: retro gressive accumulation landslide and thrust load-induced accumulation landslide, and detailed the deformation and failure characteristics of the two types of landslides, especially the deformation and failure characteristics and triggering factors(rainfall infiltration) of the thrust load-induced accumulation landslide studied in this paper(In the “2. Failure Mode of Thrust Load-Induced Accumulation Landslide Triggered by Rainfall Infiltratio” in paragraphs 1-3, Figure 1,Figure 2,Figure 14). Then, from the perspective of the material composition and spatial structure(this is an internal factor that controls the occurrence of the thrust load-induced accumulation landslide, and rainfall infiltration is an external triggering factor) of the rock-soil mass on the slope (soft surface), the spatial distribution characteristics of a two-stage composite sliding surface with steep up and slow down are summarized(Figure 1,Figure 2). Two failure modes are proposed according to whether the material composition of the sliding zone in the down-slow section and the up-steep section is the same(Figure 2), and the influence of the above two modes is considered in the derivation of the stability calculation formula(Line 284-290, Formula 33-35), and the two cases of whether the soil in up-steep section is completely weakened are also considered. The two situations(Line 284-290, Formula 33-35)( Pattern (1)+Situation (1), Pattern (1)+Situation (2), Pattern (2)+Situation (1), Pattern (2)+Situation (2)) in the above two modes(Figure 2) can generally cover all the characteristics of deformation and failure of the thrust load-induced accumulation landslide (triggered by rainfall infiltration). This is not involved in previous studies, so the author believes that this is one of the innovations of this paper.
“2) Research results on the mechanism of water–soil interactions after rainfall infiltration are not universal because some landslides have extremely unique characteristics.”
At present, there are many research results on the water-soil interaction mechanism caused by rainfall infiltration, which leads to a strong pertinence of each research result. Each landslide has its own characteristics in terms of rainfall infiltration and physical and mechanical properties of rock-soil mass, and the corresponding research results on the water-soil interaction mechanism cannot be fully applicable. Therefore, this paper only briefly describes the mechanism of how the water-soil interaction weakens the mechanical parameters of the sliding zone soil, but directly finds the relationship between the shear stiffness and the traditional mechanical parameters (Cohesion c, internal friction angle φ),use the weakening coefficient k related to the shear stiffness to replace the decrease of the mechanical parameters of the sliding zone soil. This method has two advantages: 1. It avoids the influence of multiple mechanical parameters on the calculation, and only uses one parameter for stability calculation, which improves the accuracy of the calculation; 2.This study avoids the most complex and changeable mechanism of water-soil interaction in landslide research and provides a new idea for the study of this kind of landslide.
“3) The characteristics of gradual creep failure of accumulation landslides have not been considered. 4) After rainfall infiltration, the mechanical properties of accumulation slopes exhibit gradual weakening, but the geotechnical physical parameters considered in the calculation of slope stability are all fixed values. Therefore, the spatiotemporal variability of the mechanical parameters is neglected.”
Rainfall infiltration into the interior of the slope is a gradual process. In the thrust load-induced accumulation landslides triggered by rainfall infiltration, rainfall usually infiltrates the internal sliding zone of the slope along the trailing edge of the landslide, and gradually seepage along the sliding zone to the leading edge (Figure 2,Figure 14), this leads to the weakening of the sliding zone soil is also a gradual process. In the previous studies on the thrust load-induced accumulation landslides, there are few studies on gradual creep, and most of them are qualitative studies; In terms of stability calculation, the sliding zone is usually considered as a whole and each state is given a set of fixed parameters for calculation. It is difficult to summarize the weakening characteristics of the sliding zone as a whole. Therefore, the advantages of this method are as follows: 1. Combine the current theory of gradual creep and push-type accumulation landslide, and apply it to the stability calculation formula on the basis of qualitative analysis and summary. 2. Use the weakening coefficient k related to the shear stiffness to replace the fixed value of the traditional calculation, and the weakening coefficient k is changing throughout the calculation process, to a certain extent, it reflects the weakening of the mechanical parameters of the sliding zone soil, the characteristic of periodic changes in time, and the characteristics of differences in space.
5) The traditional stability calculation method cannot fully reflect the characteristics of shear stress and displacement changes in the sliding zone.
As explained in the previous, in the previous studies on the thrust load-induced accumulation landslides, there are few studies on gradual creep, and most of them are qualitative studies; In terms of stability calculation, the sliding zone is usually considered as a whole and each state is given a set of fixed parameters for calculation. It is difficult to summarize the weakening characteristics of the sliding zone as a whole. In addition, the mechanical parameters (Cohesion c, Internal friction angle φ) given by it are usually obtained from experiments, parameter inversion, and analogy, and it is difficult to reflect the characteristics of shear stress and displacement changes. It is worth noting that the method in this paper is based on the negative exponential constitutive equation and the constitutive equation of sliding zone soil. The equation itself can effectively reflect the relationship between shear stress and shear displacement (Equation 1).
Based on the above explanation, the research ideas proposed by the author are innovative:1. First, this paper summarizes the deformation and failure characteristics of the thrust load-induced accumulation landslide. 2. Subsequently, the rock-soil composition and spatial structure characteristics of the sliding zone soil of this type of landslide are summarized, and four modes of deformation and failure of the thrust load-induced accumulation landslide are derived from this. (Pattern (1)+Situation (1), Pattern (1)+Situation (2), Pattern (2)+Situation (1), Pattern (2)+Situation (2)) 3. Then, the qualitative analysis of gradual creep is combined with the constitutive equation, and the weakening coefficient k related to the shear stiffness is used to replace the mechanical parameters of the sliding zone soil (Cohesion c, Internal friction angle φ), and applying the "S"-shaped curve to establish a state curve equation of the weakening coefficient of sliding zone soil, which can reflect the real situation of the weakening process. 4 Finally, applying the deduced results to the stability calculation, which provides a new idea for slope stability calculation research.
(4)About scientific content 2
Your opinion: “The description of the kind of landslides you are studying with your model is unclearly presented. You should be more consistent throughout the paper, using the same terms to define your landslides and the involved materials and, more importantly, you should refer to the terminolgoy of a generally accepted landslide classification (e.g. Varnes, or Hungr). ”.
Reply to this question: The author found that by consulting related landslide classification literature (including the scholars listed by the editor). Varnes (classified according to the characteristics of rock and soil mass movement) classifies slope failures into Falls, Topples, Slides, Lateral spreads, Flows, Complex. Hutchinson(classified according to the characteristics of rock and soil mass movement) classifies slope failures into Rebound、Creep、Sagging of mountain slopes、Landslides、Debris movements of flow-like form、Topples、Falls、Complex slope movements. Although their classification is generally accepted by scholars, the starting point and gist of this paper are the mechanical mechanism of deformation and failure of accumulation landslides. Therefore, according to the mechanical mechanism of landslide deformation and failure, the author divides accumulation layer landslides into two types: retro gressive accumulation landslide and thrust load-induced accumulation landslide, and made changes and additions in "2. Failure Mode of Thrust Load-Induced Accumulation Landslide Triggered by Rainfall Infiltratio" (paragraphs 1-3, Figure 1, Figure 2, Figure 14), which introduces the mechanical mechanism of these two types of landslides in detail from both theoretical and practical cases.
(5)About scientific content 3
Your opinion: “Overall, it is not clear how you choose the values of your models (degradation of material mechanical properties and evolution of the S-shaped curve) and adapt them to your case study. Back-analysis of field measurements of displacements and rainfall? Literature indications? Without clarifying this, nobody can learn from your study how to apply the presented models.”.
Reply to this question: I am very sorry for this, the author's expression in this aspect is somewhat deficient. After the last review, the author has supplemented and perfected the meaning of each parameter in the formula derivation process. If there are still parts that are difficult for readers to understand, the editor is asked to propose and the author continues to modify. In addition, regarding the parameter assignment problem in the case analysis, the author has added this point in the paper (Line 351-368), among which: 1). ” and were mainly obtained from triaxial test results of sliding zone soil. In the absence of a triaxial test, they can also be indirectly obtained by direct shear test combined with the Duncan-Zhang bicurve model”. The author has listed the references of the solution method. The and in this paper are calculated indirectly according to this method. 2). “The values of , , , and are mainly based on the softening coefficient of sliding zone soil”. These parameters are based on on-site rock and soil mechanics test results, and calculated by comprehensively analyzing the drop range of mechanical parameters from natural moisture content to saturated moisture content(Cohesion c, internal friction angle φ), and get the result. 3). and are empirical values obtained based on the existing sensitivity analysis results(by consulting the literature "[59]"). 4). is based on on-site rock and soil mechanics test results, and combined with existing research results give value for it (combine the literature "[43,59]" to obtain comprehensive values). 5). ρ1 and ρ2 are obtained based on on-site rock and soil physical test results. 6). The remaining parameters , , , , , , , , are calculated based on the parameters actually measured on-site and combined with the geological generalization model(Figure 9).
(6)Regarding the author’s thanks to the editor and reviewers
After two rounds of review, the quality of this paper has been improved due to the editor's suggestions for revision. For this, the author expresses my sincere thanks! There are a total of three reviewers in the first round. Among them, the decisions of Reviewer#1 and Reviewer#2 were major revisions, and the decision of Reviewer#3 was a slight revision. After the first round of revisions, the author was approved by Reviewer#2. I hope the editor can carefully review the paper again this time, and the new opinions put forward will continue to improve the quality of the paper, and the author will continue to actively cooperate with the editor and reviewers. The author apologizes again for the poor English expression before!
I hope that mankind can overcome COVID-19, long live the friendship between China and Serbia people!
Kind regards,
Dr. Zhou Zhou
State Key Laboratory of Geohazard Prevention and Geoenvironment Protection, Chengdu University of Technology

This manuscript is a resubmission of an earlier submission. The following is a list of the peer review reports and author responses from that submission.